# Predictive performance of oximetry in detecting sleep apnea in surgical patients with cardiovascular risk factors

**Rida Waseem[1], Matthew T. V. Chan[2], Chew Yin Wang[3], Edwin Seet[4], Frances Chung[1]\***

**1** Toronto Western Hospital, University Health Network, University of Toronto, Toronto, Ontario, Canada, **2** The Chinese University of Hong Kong, Hong Kong Special Administrative Region, China, **3** University of Malaya, Kuala Lumpur, Malaysia, **4** Khoo Teck Puat Hospital, National Healthcare Group, Singapore, Singapore

\* frances.chung@uhn.ca

## Abstract

**Data Availability Statement:** All relevant data are within the manuscript and its Supporting information files.

### Introduction

In adults with cardiovascular risk factors undergoing major noncardiac surgery, unrecognized obstructive sleep apnea (OSA) was associated with postoperative cardiovascular complications. There is a need for an easy and accessible home device in predicting sleep apnea. The objective of the study is to determine the predictive performance of the overnight pulse oximetry in predicting OSA in at-risk surgical patients.

### Methods

This was a planned post-hoc analysis of multicenter prospective cohort study involving 1,218 at-risk surgical patients without prior diagnosis of sleep apnea. All patients underwent home sleep apnea testing (ApneaLink Plus, ResMed) simultaneously with pulse oximetry (PULSOX-300i, Konica Minolta Sensing, Inc). The predictive performance of the 4% oxygen desaturation index (ODI) versus apnea-hypopnea index (AHI) were determined.

### Results

Of 1,218 patients, the mean age was 67.2 ± 9.2 years and body mass index (BMI) was 27.0 ± 5.3 kg/m$^2$. The optimal cut-off for predicting moderate-to-severe and severe OSA was ODI ≥15 events/hour. For predicting moderate-to-severe OSA (AHI ≥15), the sensitivity and specificity of ODI ≥ 15 events per hour were 88.4% (95% confidence interval [CI], 85.7–90.6) and 95.4% (95% CI, 94.2–96.4). For severe OSA (AHI ≥30), the sensitivity and specificity were 97.2% (95% CI, 92.7–99.1) and 78.8% (95% CI, 78.2–79.0). The area under the curve (AUC) for moderate-to-severe and severe OSA was 0.983 (95% CI, 0.977–0.988) and 0.979 (95% CI, 0.97–0.909) respectively.

**Funding:** The study was funded through grants from the Health and Medical Research Fund (09100351), Hong Kong, National Healthcare Group-Khoo Teck Puat Hospital, Small Innovative Grants (12019, 15201), University Health Network Foundation (Ontario, Canada), University of Malaya, High Impact Research Grant (UM.C/625/1/HIR/067), Malaysian Society of Anaesthesiologists K Inbasegaran Research Grant, and Auckland Medical Research Foundation, New Zealand. ResMed has supplied the ApneaLink devices and PULSOX-300i oximeter wristwatch in all sites as an unrestricted loan. These were returned at the end of the study. Role of the Funder/Sponsor: The study funders/sponsors had no role in the design and conduct of the study; collection, management, analysis, and interpretation of the data; preparation, review, or approval of the manuscript; and decision to submit the manuscript for publication.

**Competing interests:** Dr Chung reported receiving grants from the Ontario Ministry of Health and Long-term Care, University Health Network Foundation. STOP-Bang questionnaire: proprietary to University Health Network. No other authors reported disclosures. This does not alter our adherence to PLOS ONE policies on sharing data and materials.

**Abbreviations:** AHI, Apnea-hypopnea Index; BMI, Body Mass Index; HSAT, Home sleep apnea testing; ODI, oxygen desaturation index; OSA, Obstructive sleep apnea; PSG, polysomnography.

## Discussion

ODI from oximetry is sensitive and specific in predicting moderate-to-severe or severe OSA in at-risk surgical population. It provides an easy, accurate, and accessible tool for at-risk surgical patients with suspected OSA.

## Introduction

Obstructive sleep apnea (OSA) is a common sleep-disordered breathing, affecting. nearly one billion people worldwide, of whom 425 million has moderate-to-severe OSA disease [1]. The prevalence of moderate-to-severe OSA was reported to be 6–17% in the general population [2]. OSA is often undiagnosed and is associated with mortality and morbidity including hypertension, cardiovascular diseases, and neurocognitive impairment [1]. Among the surgical patients, OSA is associated with increased complications and adverse outcomes [3, 4]. In adults with cardiovascular risk factors undergoing major noncardiac surgery, unrecognized severe OSA was significantly associated with higher risk of 30-day postoperative cardiovascular complications [5]. Thus, it is important to have an early screening, diagnosis, and treatment of OSA in at-risk surgical patient.

The laboratory polysomnography (lab-PSG) is the gold standard for diagnosing OSA, but there are limitations due to its cost and accessibility. Home sleep apnea testing (HSAT) is an alternative option and several studies have validated the accuracy of HSAT in predicting sleep apnea with lab-PSG [6–8]. Nevertheless, both lab-PSG and HSAT require substantial expertise of sleep medicine specialists for interpretation.

Overnight pulse oximetry is an accessible and economical tool for screening OSA. Oximetry has been validated to screen patients against apnea-hypopnea index (AHI) from lab-PSG and portable devices. These studies are mostly limited to sleep clinic patients from a single centre [9–13]. The predictive performance of the overnight oximetry is not known for at-risk surgical patients undergoing major non-cardiac surgery. The objective of the study is to examine the predictive performance of overnight oximetry versus HSAT (ApneaLink Plus; ResMed, San Diego, CA) in predicting OSA in at-risk surgical population. We hypothesize that oxygen desaturation index (ODI) from overnight oximetry in comparison to AHI from HSAT would show high sensitivity and specificity in predicting OSA.

## Methods

### Study design

This was a planned, post hoc analysis of a multicenter, prospective cohort study of patients having major noncardiac surgery [5]. Data were collected in five countries at eight hospitals from January 2012 to July 2017. Ethics approval was obtained by all participating institutions. Name of the institutional review board or ethics committee that approved the study are as follows,1. The Chinese University of Hong Kong, Prince of Wales Hospital, Hong Kong, 2. Tuen Mun hospital, Hong Kong, 3. University Malaya Medical Centre, Kuala Lumpur, Malaysia, 4. Hospital Kuala Lumpur, Malaysia, 5. Khoo Teck Puat Hospital, Singapore, Singapore, 6. Scarborough Health Network-Central Campus, Ontario, Canada, 7. Middlemore Hospital, Manukau City, New Zealand, and 8. Auckland City Hospital, Auckland, New Zealand.

All patients voluntary provided written informed consent to the study (ClinicalTrials.gov Identifier: NCT01494181). The study complied with the Declaration of Helsinki. The baseline characteristics and comorbidities of the patients were recorded before surgery.

## Participants

Patients undergoing major elective non-cardiac surgery were approached for the study. The following are the inclusion criteria of the study: 1) age $\geq$45 year undergoing major noncardiac surgery (intraperitoneal, major orthopedic, or vascular) and 2) had 1 or more risk factors for postoperative cardiovascular events (i.e. history of coronary artery disease, heart failure, stroke or transient ischemic attack, diabetes requiring treatment, and renal impairment with preoperative plasma creatinine concentration >175 μmol/L). The exclusion criteria were: 1) prior diagnosis or undergoing corrective surgery for OSA and 2) patients requiring greater than two days of mechanical lung ventilation post-surgery [5].

## Home sleep apnea testing (HSAT) and pulse oximetry

All patients underwent a preoperative overnight sleep study at home or in the hospital using a type 3 HSAT (ApneaLink Plus; ResMed, San Diego, CA). It includes a nasal pressure transducer which measure flow limitation and snoring, and records on a 16-bit signal processor and a sampling rate of 100 Hz. In addition to HSAT, oxyhemoglobin saturation was simultaneously monitored using high-resolution pulse oximetry wristwatch (PULSOX-300i, Konica Minolta Sensing, Inc, Osaka, Japan). The oxygen probe of the oximetry was attached to finger of the nondominant hand. The sampling frequency was set as 1 Hz with an averaging time of 3s. The resolution was 0.1%. All patients breathed room air during recording.

The data was extracted the next morning using ApneaLink and Profox (Profox Associates, Escondido, California, USA) software, respectively. All data was processed by a technician blinded to the clinical data and the STOP-Bang score. The sleep parameters variables were extracted from the ApneaLink Plus including the AHI. The sleep-associated apnea and hypopnea events were scored according to the American Academy of Sleep Medicine criteria. Apnea was defined as airflow reduction of $\geq$90% for $\geq$10s from baseline. Hypopnea was defined as reduction in airflow for $\geq$30% for $\geq$10s from baseline and associated with $\geq$3% oxyhemoglobin desaturation [14]. Patients with an AHI $\geq$15 events per hour were considered to have moderate-to-severe OSA, and those with AHI $\geq$30 events per hour were considered to have severe OSA.

ODI, duration oxygen saturation ($SpO_2$) <90%, lowest, and average $SpO_2$ were extracted from oximetry using Profox software. ODI is defined as the number of events per hour with at least 4% decrease in saturation from the average saturation in the preceding 120s for at least 10s [10]. The oximetry recording data were processed which was recorded between 00:00 to 6:00 hours of night, although it was not known if patients were asleep during this entire period.

## Statistical analysis

Data was analyzed using Stata/SE 14.2 (StataCorp, College Station, TX). Demographics, oximetry, and sleep parameters were presented using descriptive statistics. Continuous data were presented using mean (standard deviation) or median (interquartile range) and categorical data were presented using frequencies (percentage), as appropriate. The relationship between the AHI and ODI was examined using Spearman correlation and agreement was displayed by the Bland Altman plot. The predictive performance of ODI at apnea-hypopnea index (AHI) (moderate-to-severe and severe OSA) cut-offs was expressed as sensitivity, specificity, positive predictive value (PPV), negative predictive value (NPV), positive likelihood ratio (LR), negative LR, accuracy, and area under curve (AUC) with 95% confidence interval (CI). The best cut-off for ODI were chosen by optimal parameters. A P value <0.05 was considered statistically significant. The sample size was based on the primary outcome of the study [5].

## Results

In the POSA study, a total of 1,364 patients were recruited, 1,218 patients were included for analysis, and 146 patients were excluded because of cancelled surgery, failure of sleep study, or duration of study less than 4 hours. Demographic data, sleep parameters and comorbid conditions are shown in the Table 1. The mean age was $67.2 \pm 9.3$ years with body mass index (BMI) $27.0 \pm 5.3$ kg/m$^2$ and 60% were male. The patients were in the following ethnic groups, 666 (54.7%) Chinese, 161 (13.2%) Indian, 195 (16%) Malay, 183 (15%) Caucasian, and 13 (1.1%) others (Arabs or Black). Eighty-five percent of patients had hypertension and seventy-seven percent had diabetes (Table 1).

Using AHI, 68% of patients had OSA (AHI $\geq$5), 30% patients had moderate-to-severe OSA (AHI $\geq$15), and 12% had severe OSA (AHI $\geq$30 events per hour) (Table 2). Compared with ODI <15, patients with ODI $\geq$15 had higher median AHI (P<0.001), apnea index (P<0.001), obstructive apnea index (P<0.001), central apnea index (P<0.001), mixed apnea index (P<0.001), and hypopnea index (P<0.001) (Table 2).

To examine the association between AHI and ODI, Spearman coefficient showed that there was a very strong correlation between ODI and AHI (r = 0.907, P<0.001). The Bland-Altman plot also showed the relationship between AHI and ODI (Fig 1). The plot showed good agreement between AHI and ODI with mean difference of -0.15 ± 5.82.

**Table 1. Characteristics of patients.**

| Characteristics | N = 1,218 |
|---|---|
| **Body mass index, kg/m$^2$** | 27.0 ± 5.3 |
| **Age, years** | 67.2 ± 9.3 |
| **Male sex, n (%)** | 728 (59.8) |
| **Neck circumference, cm** | 38.5 ± 4.0 |
| **Waist circumference, cm** | 93.0 ± 12.1 |
| **Epworth sleepiness scale** | 5.1 ± 4.0 |
| **Ethnicity, n (%)** | |
| Chinese | 666 (54.7) |
| Indian | 161 (13.2) |
| Malay | 195 (16.0) |
| Caucasian | 183 (15.0) |
| Others | 13 (1.1) |
| **Comorbidities, n (%)** | |
| Hypertension | 1037 (85.1) |
| Diabetes | 938 (77.0) |
| COPD | 60 (4.9) |
| Asthma | 72 (5.9) |
| **Types of Surgery, n (%)** | |
| **Intraperitoneal** | 427 (35.1) |
| **Orthopedic** | 364 (29.9) |
| **Vascular** | 167 (13.7) |
| **Cancer** | 512 (42.0) |
| **Minimally invasive** | 345 (28.3) |
| **Other** | 260 (21.3) |

Continuous variables are expressed as mean± standard deviation as appropriate and categorical variables were presented as frequency (percentage).

COPD, chronic obstructive pulmonary disease.

**Table 2. Sleep study parameters of patients.**

| Sleep study parameters | N = 1,218 | ODI <15 | ODI ≥15 | P |
|---|---|---|---|---|
| | | N = 851 | N = 367 | |
| Apnea-hypopnea index ≥5, n (%) | 823 (67.6) | 456 (55.4) | 367 (44.6) | <0.001 |
| Apnea-hypopnea index ≥15, n (%) | 371 (30.5) | 43 (11.6) | 328 (88.4) | <0.001 |
| Apnea-hypopnea index ≥30, n (%) | 143 (11.7) | 4 (2.8) | 139 (97.2) | <0.001 |
| Duration of sleep study, hour | 9.4± 2.3 | 9.4 ± 2.0 | 9.2 ± 2.0 | 0.034 |
| Apnea-hypopnea index, events/hour | 8 (3–17) | 5 (2–8) | 25 (17–36) | <0.001 |
| Apnea index, events/hour | 2 (1–8) | 1 (0–3.3) | 12 (6–22) | <0.001 |
| Obstructive apnea index, events/hour | 1 (0–6) | 1 (0–2) | 8 (3–16) | <0.001 |
| Hypopnea index, events/hour | 4 (1–9) | 2 (1–5) | 11 (6–16) | <0.001 |
| Central apnea index, events/hour | 0 (0–1) | 0 (0–0) | 1 (0–2) | <0.001 |

Continuous variables are expressed as mean± standard deviation or median (interquartile) as appropriate and categorical variables were presented as frequency (percentage).

Apnea index is the number of apnea events per hour of recording with airflow reduction of ≥ 90% from baseline for ≥ 10 seconds. Hypopnea index is the number of hypopnea events per hour of recording with airflow reduction ≥ 30% from baseline for ≥ 10 seconds and associated with ≥ 3% oxyhemoglobin desaturation. ODI, oxygen desaturation index.

The predictive parameters were estimated for four ODI cut-offs (ODI ≥5, ≥10, ≥15 and ≥30 events per hour) against AHI ≥15 and ≥30 (Table 3). The result showed that ODI ≥15 had optimal sensitivity and specificity in predicting moderate-to-severe OSA and severe OSA. To predict moderate-to-severe OSA, the sensitivity of ODI ≥15 was 88.4% (95% CI, 85.7–90.6) and the specificity was 95.4% (95% CI, 94.2–96.4), while PPV and NPV were 89.4% (95% CI, 86.7–91.6) and 94.9% (95% CI, 93.8–95.9) respectively. The area under the curve (AUC) was 0.983 (95% CI, 0.977–0.988) (Fig 2A). To predict severe OSA, the sensitivity and specificity of ODI ≥15 were 97.2% (95% CI, 92.7–99.1) and 78.8% (95% CI, 78.2–79.0), while PPV and NPV were 37.9% (95% CI, 36.1–38.6) and 99.5% (95% CI, 98.8–99.8) respectively. The AUC was 0.979 (95% CI, 0.970–0.909) (Fig 2B).

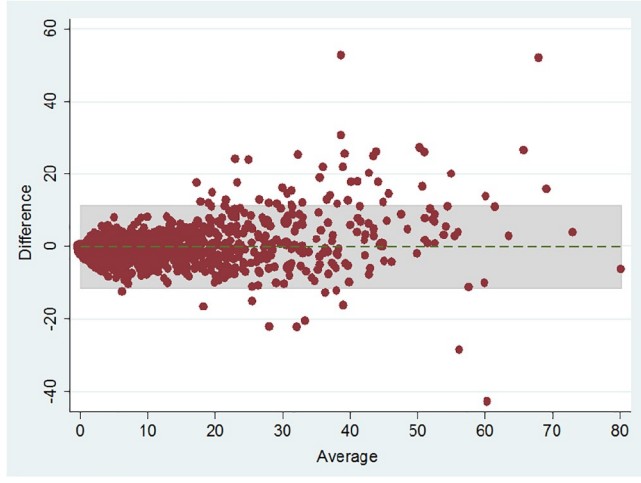

**Fig 1. Bland-Altman plot of apnea-hypopnea index (AHI) from portable device versus oxygen desaturation index (ODI) from simultaneously recorded oximetry (N = 1,218).** The vertical axis represents difference of AHI and ODI and horizontal axis represents average of AHI and ODI. Shaded area represents limits of agreement (±2 SD) while dotted line represents mean difference.

**Table 3. Predictive parameters of oxygen desaturation index (ODI) at different cut-offs of apnea-hypopnea index (AHI).**

| N = 1,218 | ODI ≥5 | ODI ≥10 | ODI ≥15 | ODI ≥30 |
|---|---|---|---|---|
| **AHI ≥15 AUC: 0.983 (0.977–0.988)** | | | | |
| **Sensitivity, %** | 100 (98.7–100) | 99.2 (97.5–99.8) | 88.4 (85.7–90.6) | 32.6 (31.4–32.6) |
| **Specificity, %** | 38.7 (38.2–38.7) | 78.2 (77.4–78.4) | 95.4 (94.2–96.4) | 100 (95.5–100) |
| **PPV, %** | 41.7 (41.2–41.7) | 66.5 (65.4–66.9) | 89.4 (86.7–91.6) | 100 (96.2–100) |
| **NPV, %** | 100 (98.6–100) | 99.5 (98.6–99.9) | 94.9 (93.8–95.9) | 77.2 (76.8–77.2) |
| **LR+** | 1.63 (1.60–1.63) | 4.5 (4.3–4.6) | 19.2 (14.9–24.9) | Inf |
| **LR-** | 0 (0–0.03) | 0.01 (0–0.03) | 0.12 (0.09–0.15) | 0.67 (0.67–0.69) |
| **Accuracy, %** | 57.4 (56.6–57.4) | 84.6 (83.5–84.9) | 93.3 (91.6–94.6) | 79.5 (78.7–79.5) |
| **AHI ≥30 AUC: 0.979 (0.970–0.988)** | | | | |
| **Sensitivity, %** | 100 (96.8–100) | 100 (96.8–100) | 97.2 (92.7–99.1) | 74.1 (68.9–78.0) |
| **Specificity, %** | 30.5 (30.1–30.5) | 61.9 (61.4–61.9) | 78.8 (78.2–79.0) | 98.6 (97.9–99.1) |
| **PPV, %** | 16.1 (15.6–16.1) | 25.9 (25.0–25.9) | 37.9 (36.1–38.6) | 87.6 (81.4–92.2) |
| **NPV, %** | 100 (98.6–100) | 100 (99.3–100) | 99.5 (98.8–99.8) | 96.6 (95.9–97.1) |
| **LR+** | 1.44 (1.38–1.44) | 2.62 (2.51–2.62) | 4.58 (4.25–4.73) | 53.1 (32.9–89.2) |
| **LR-** | 0 (0–0.11) | 0 (0–0.05) | 0.04 (0.01–0.09) | 0.26 (0.22–0.32) |
| **Accuracy, %** | 38.7 (37.9–38.7) | 66.3 (65.6–66.3) | 81.0 (79.9–81.4) | 95.7 (94.5–96.6) |

AHI, apnea hypopnea index, ODI, oxygen desaturation index, AUC, area under curve; PPV, positive predictive value; NPV, negative predictive value; AHI, apnea-hypopnea index; LR, likelihood ratio.

## Discussion

In surgical patients with cardiovascular risk factors, we showed a strong association between ODI from overnight oximetry and AHI from HSAT. Based on the optimal sensitivity and specificity, we identified that ODI ≥15 is useful to accurately identify moderate-to-severe and severe OSA in surgical patients with cardiovascular risk factors. To predict moderate-to-severe OSA, ODI ≥15 events per hour showed high accuracy of 93.3% and AUC of 0.98. Similarly, for severe OSA, ODI ≥15 events per hour showed very good predictive performance as shown in the results. Ideally, identification of patients with OSA, and those with suspected OSA should take place well in advance of elective surgery to allow time for potential evaluation and management of OSA preoperatively [15]. In clinical practice, many patients are identified close to the operative time, often just days before surgery. Oximetry serves as a simple tool to identify these at-risk surgical patients to ensure optimal perioperative management.

A recent systematic review on patients referred to sleep clinic also recommended using ODI ≥15 events per hour for predicting OSA but ODI ≥10 events per hour for further evaluation of OSA [9]. Although there has been a number of studies to predict OSA using oximetry [10, 11, 16, 17], this is the first study to show that overnight pulse oximetry is a valid screening tool in predicting OSA in surgical patients with cardiovascular risk factors. Additionally, this multicenter study involved a large sample size of different ethnic groups, which allows the findings to be generalizable to diverse population.

Screening for OSA in the perioperative setting has been recommended, as OSA creates a challenge for surgeons and anesthesiologists due to increased adverse outcomes [3, 18]. The combinations of opioids plus sedatives were associated with worse outcomes sustained by OSA patients after a critical event [3]. Before surgery, identifying patients at high risk for OSA with a screening tool such as the STOP-Bang questionnaire [19], for targeted perioperative precautions and interventions may help to reduce patient complications. For those

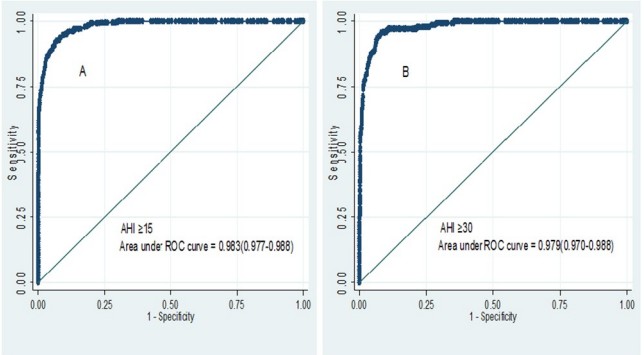

**Fig 2. Receiver operating curve for oxygen desaturation index (A and B) at AHI ≥15 and AHI ≥30 events per hour respectively.** Abbreviations: AHI, apnea-hypopnea index.

who were identified to be at high risk of OSA with co-morbid cardiovascular risk factors, we showed that oximetry serves as a quick and accessible tool to predict OSA. Compared to in-lab PSG which is not economical to use in resource limited situations and often delays treatment by long wait time [20]. Our study showed that high predictive performance of oximetry using ODI ≥15 and ODI ≥30 for predicting moderate-to-severe OSA and severe OSA. Regardless, ODI ≥15 cannot distinguish between moderate-to-severe or severe OSA. In the clinical setting if someone has ODI ≥15, they can be considered as high risk of OSA. This helps in further management of high-risk patients. However, PSG has been recommended to evaluate patients with other sleep disorders and comorbid conditions [21, 22]. Although PSG provides more accurate information compared to HSAT as well as oximetry, lack of accessibility has limited its use in preoperative assessments. Therefore, oximetry is valuable in identifying at risk patients for optimal perioperative management. After surgery, we can refer patients with ODI ≥15 or greater for further evaluation and management [15].

The recent coronavirus disease (COVID-19) pandemic has been shaping the mode of health care delivery, and telemedicine is particularly beneficial for patient assessment before surgery. Due to the highly contagious nature of the COVID-19, individuals are asked to limit unnecessary physical interaction and maintain physical distance. In such situation, oximetry is a viable alternative for in-lab polysomnography as well as HSAT which requires more equipment and instruction. Oximetry is a simple device which does not require extensive patient instructions and is mailed to patients' home. It is easily disinfected due to its size.

## Limitations

Our study includes a large representative sample of international patients from eight clinical sites in five countries which support the generalizability of our results. This study has some limitations. The inclusion criteria of the study involved surgical patients with at least one cardiovascular risk factor. The results may not be applicable to patients without cardiovascular risk factors. Nevertheless, the prevalence of cardiovascular factors in the older populations is around 75% [23]. Thus, our results involving surgical patients with cardiovascular risk factors may be applicable to the older general population. Another limitation was that we use type 3 home sleep apnea testing instead of in-laboratory polysomnography, but recent studies showed comparable performance of type 3 sleep apnea device with PSG when predicting moderate-to-severe OSA [6, 24].

## Conclusion

In patients with cardiovascular risk factors undergoing major non-cardiac surgery, we found that ODI ≥15 events per hour showed a high accuracy to predict moderate-to-severe OSA and severe OSA. Overnight pulse oximetry serves as an easy-to-use, low cost, and accessible tool for the identification of OSA. The accessibility of oximetry makes it especially valuable in COVID-19 era, which demands a contactless tool. The quick screening by oximetry will help clinicians in urgent screening and management of OSA, which is often delayed by long waiting time of PSG.

## Supporting information

**S1 Data.**
(XLSX)

## Acknowledgments

The authors acknowledged the special assistance of Hou Yee Lai, MBBS; Eleanor F. F. Chew, MBBS; Benny C. P. Cheng, MBBS; Timothy G. Short, MD in data collection.

 **Group Information**: The Postoperative Vascular Complications in Unrecognized Obstructive Sleep Apnea (POSA) Study Investigators: **Steering Committee**: Frances Chung (chair); Matthew Chan, Chew-Yin Wang, Edwin Seet. **Investigators: Canada**: *Scarborough Health Network—Central Campus*: Frances Chung, MBBS, Stanley Tam, MD, Sohail Iqbal, BSc; **Hong Kong**: *Prince of Wales Hospital*: Matthew Chan, MBBS, PhD, Gordon Choi, MBBS, David Hui, MD, Tony Gin, MD, Matthew Tsang, BSc, Beaker Fung, BSc, Angela Miu, BSc, Alex Lee, MSc; *Tuen Mun Hospital*: Benny Cheng, MBBS, Carmen Lam, MBBS, Sharon Tsang, MBChB, PhD, Chuen Ho Cheung, MBChB, Hoi Lam Pang, MBBS; **Malaysia**: *University of Malaya Medical Centre*: Chew Yin Wang, MBChB, Hou Yee Lai, MBBS, Carolyn C.W. Yim, MBBS, Alvin S.B. Tan, MBBS, Ching Yen Chong, BA, Jason H. Kueh, BSc, Xue Lin Chan, MD; *Hospital Kuala Lumpur*: Eleanor F.F. Chew, MBBS, Su Yin Loo, MD, Simon M.T. Hui, MBBS; **New Zealand**: *Middlemore Hospital*: Joyce Tai, MBChB, Stuart Walker, MBBS, Sue Olliff, BSc; *Auckland City Hospital*: Ivan Bergman, MBBS, Nicola Broadbent, MBBS, Maartje Tulp, MBBS, Timothy Short, MD, Davina McAllister, BSc; **Singapore**: *Khoo Teck Puat Hospital*: Edwin Seet, MBBS, Pei Fen Teoh, MBBS, Audris Chia, BSc.

## Author Contributions

**Conceptualization:** Rida Waseem, Frances Chung.

**Data curation:** Matthew T. V. Chan, Chew Yin Wang, Edwin Seet, Frances Chung.

**Formal analysis:** Rida Waseem.

**Funding acquisition:** Matthew T. V. Chan, Chew Yin Wang, Edwin Seet, Frances Chung.

**Investigation:** Frances Chung.

**Methodology:** Rida Waseem.

**Resources:** Matthew T. V. Chan, Chew Yin Wang, Edwin Seet, Frances Chung.

**Supervision:** Frances Chung.

**Validation:** Matthew T. V. Chan, Frances Chung.

**Writing – original draft:** Rida Waseem.

**Writing – review & editing:** Rida Waseem, Matthew T. V. Chan, Chew Yin Wang, Edwin Seet, Frances Chung.

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
