## [Decision Letter · Decision Letter 0]

23 Dec 2020

PONE-D-20-33011

Predictive Performance of Oximetry in Diagnosis of Sleep Apnea in Surgical Patients with Cardiovascular Risk Factors

PLOS ONE

Dear Dr. Chung,

Thank you for submitting your manuscript to PLOS ONE. After careful consideration, we feel that it has merit but does not fully meet PLOS ONE’s publication criteria as it currently stands. Therefore, we invite you to submit a revised version of the manuscript that addresses the points raised during the review process.

Please note that I had a difficult time finding a second reviewer for this manuscript. Therefore, I served as the second reviewer. My comments are in the Editor's comments below.  The issue of the relative importance of the findings is a key one; I highly encourage the authors to find a way to include outcomes data when they revise the manuscript. 

We look forward to receiving your revised manuscript.

Kind regards,

James Andrew Rowley

Academic Editor

PLOS ONE

Journal Requirements:

2. Please provide additional details regarding participant consent. In the ethics statement in the Methods and online submission information, please ensure that you have specified whether consent was informed.

3. In your Methods section, please provide additional information about the participant recruitment method and the demographic details of your participants. Please ensure you have provided sufficient details to replicate the analyses such as: a) a description of how participants were recruited, and b) descriptions of where participants were recruited and where the research took place.

4.Thank you for including your ethics statement: 

"Ethics approval was obtained from each participating institution. Patients provided written consent to the study.".   

5.Thank you for stating the following in the Competing Interests section:

"Conflict of Interest Disclosures: Dr Chung reported receiving grants from the Ontario Ministry of Health and Long-term Care, University Health Network Foundation. STOP-Bang questionnaire: proprietary to University Health Network. No other authors reported disclosures. "

6.We note that you have indicated that data from this study are available upon request. PLOS only allows data to be available upon request if there are legal or ethical restrictions on sharing data publicly. For information on unacceptable data access restrictions, please see http://journals.plos.org/plosone/s/data-availability#loc-unacceptable-data-access-restrictions.

Additional Editor Comments (if provided):

In this manuscript, the authors compare the performance of pulse oximetry for the diagnosis of OSA compared to a Type 3 device (ApneaLink) in a cohort of surgical patients with cardiovascular risk factors. They found good correlation as measured by the Spearman correlation and agreement by Bland-Altman analysis.

My major comment is that it is unclear what this study to the literature. Yes, the cohort is one of the largest to be studied, was multi-center and included patients with CV risk factors. But in the end, it’s just another study showing that a good oximeter providing an ODI can correlate with a Type 3 device. Thus, the authors need to convince me that there is some truly unique about this data set. Therefore, what would tremendously help is outcomes data. I am hoping that since the dataset is from a large cohort study of surgery, that the authors would have post-surgical outcomes, either cardiac or pulmonary (or ideally both). The authors could then determine if ODI alone (ie, not in conjunction with the AHI) is predictive of post-surgical complications. Such an analysis would greatly strengthen the data and provide more than just another validation of oximetry v. HSAT.

Minor comments

1. The authors included patients with CV risk factors but unclear if pulmonary risk factors were an inclusion or exclusion. Key as COPD is a risk factor for CV disease.

2. Why did the authors exclude patients requiring mechanical ventilation > 2 days? Given my question about outcomes above, very key to include ALL patients in such an analysis. Does this group have pre-surgery HSAT/ODI data?

3. Results: the authors present data in two different ways: either to no decimal place or to the 10th decimal place; should be uniform in both tables and would suggest to the 10th decimal place.

4. Table 2: what is respiratory event index?

5. Page 14: the paragraph on COVID seems to be added just because we’re in the midst of the pandemic. Not sure needed for the greater picture of this study.

6. While it appears that ODI>15 is good for both moderate-severe and severe AHI cutoff, in reality, if use ODI>15, will not be able to distinguish between the two. This should be clarified.

Reviewers' comments:

Reviewer's Responses to Questions

**Comments to the Author**

1. Is the manuscript technically sound, and do the data support the conclusions?

Reviewer #1: Yes

2. Has the statistical analysis been performed appropriately and rigorously? 

Reviewer #1: Yes

3. Have the authors made all data underlying the findings in their manuscript fully available?

Reviewer #1: No

4. Is the manuscript presented in an intelligible fashion and written in standard English?

Reviewer #1: Yes

5. Review Comments to the Author

Reviewer #1: Thank you for this well-written and excellent manuscript on HSAT vs pulse-oximetry in a preoperative population. This is an important subject considering the increasingly difficult access of preoperative PSG services. I only have a few minor questions and issues.

My primary concern is mostly a wordsmithing concern. Language throughout the manuscript refers to the 'diagnostic' performance of ODI from pulse oximetry. This is innacurate, as OSA cannot be diagnosed solely from oxygen desaturation -- for instance, some of these patients may have been found to have primary central sleep apnea. I would prefer language throughout the manuscript be revised to reflect 'predictive' performance of ODI for OSA. A few examples:

"There is a need for an easy and accessible home device in diagnosing sleep apnea." This exists in the form of HSAT

"The objective of the study is to determine the diagnostic performance of the overnight pulse oximetry in predicting OSA in at-risk surgical patients." Should read "predictive performance."

"high diagnostic accuracy of 93.3% and AUC of 0.98. Similarly, for severe OSA, ODI ≥15 events per hour showed very good diagnostic performance." Should revise all to read 'predictive,' not diagnostic.

My biggest methodological question is that you report hypopnea was defined as reduction in airflow for ≥30% for ≥10s from baseline. You also state you used AASM criteria for reporting AHI. Does this mean you used a 3% oxygen desaturation for hypopneas? Please specify.

A few minor methodological questions from the Methods:

1. "1,218 at-risk surgical patients without prior risk of sleep apnea." Do you mean patients without prior diagnosis here?

2. "All data was processed by a technician blinded to the study." There is only a single cohort in this study. What exactly does this statement mean? What was the technician blinded to?

6. PLOS authors have the option to publish the peer review history of their article (what does this mean?). If published, this will include your full peer review and any attached files.

Reviewer #1: No

---

## [Author Response · Author response to Decision Letter 0]

6 Apr 2021

Please see attached 'response to reviewers'.

---

## [Decision Letter · Decision Letter 1]

14 Apr 2021

Predictive Performance of Oximetry in Detecting Sleep Apnea in Surgical Patients with Cardiovascular Risk Factors

PONE-D-20-33011R1

Dear Dr. Chung,

We’re pleased to inform you that your manuscript has been judged scientifically suitable for publication and will be formally accepted for publication once it meets all outstanding technical requirements.

Kind regards,

James Andrew Rowley

Academic Editor

PLOS ONE

Additional Editor Comments (optional):

Reviewers' comments:

Reviewer's Responses to Questions

**Comments to the Author**

1. If the authors have adequately addressed your comments raised in a previous round of review and you feel that this manuscript is now acceptable for publication, you may indicate that here to bypass the “Comments to the Author” section, enter your conflict of interest statement in the “Confidential to Editor” section, and submit your "Accept" recommendation.

Reviewer #1: All comments have been addressed

2. Is the manuscript technically sound, and do the data support the conclusions?

Reviewer #1: (No Response)

3. Has the statistical analysis been performed appropriately and rigorously? 

Reviewer #1: (No Response)

4. Have the authors made all data underlying the findings in their manuscript fully available?

Reviewer #1: Yes

5. Is the manuscript presented in an intelligible fashion and written in standard English?

Reviewer #1: (No Response)

6. Review Comments to the Author

Reviewer #1: (No Response)

7. PLOS authors have the option to publish the peer review history of their article (what does this mean?). If published, this will include your full peer review and any attached files.

Reviewer #1: No

---

## [Editor Report · Acceptance letter]

27 Apr 2021

PONE-D-20-33011R1 

Predictive Performance of Oximetry in Detecting Sleep Apnea in Surgical Patients with Cardiovascular Risk Factors 

Dear Dr. Chung:

I'm pleased to inform you that your manuscript has been deemed suitable for publication in PLOS ONE. Congratulations! Your manuscript is now with our production department. 

Kind regards, 

on behalf of

Dr. James Andrew Rowley 

Academic Editor

PLOS ONE